# Molehill Mountain feasibility study: Protocol for a non-randomised pilot trial of a novel app-based anxiety intervention for autistic people

Bethany Oakley[1]*, Charlotte Boatman[1,2], Sophie Doswell[3], Antonia Dittner[3], Andrew Clarke[4], Ann Ozsivadjian[2], Rachel Kent[5], Adrian Judd[6], Saffron Baldoza[6], Amy Hearn[6], Declan Murphy[1,7,8], Emily Simonoff[2,8], The Molehill Mountain Advisory Group[¶]

1 Department of Forensic and Neurodevelopmental Sciences, Institute of Psychiatry, Psychology and Neuroscience, King's College London, Camberwell, London, United Kingdom, 2 Department of Child and Adolescent Psychiatry, Institute of Psychiatry, Psychology and Neuroscience, Camberwell, London, United Kingdom, 3 National Adult ADHD and ASD Psychology Service (NAAAPS), Behavioural & Developmental Psychiatry, Monks Orchard House, Bethlem Royal Hospital, Beckenham, Kent, United Kingdom, 4 Autistica, Suite B, London, United Kingdom, 5 Michael Rutter Centre for Children & Young People, Maudsley Hospital, London, United Kingdom, 6 Molehill Mountain Advisory Group, 7 Sackler Institute for Translational Neurodevelopment, King's College London, Denmark Hill, London, United Kingdom, 8 South London and Maudsley NHS Foundation Trust, London, United Kingdom

¶ Membership of the Molehill Mountain advisory group is provided in the acknowledgments.
* bethany.oakley@kcl.ac.uk

**Data Availability Statement:** No datasets were generated or analysed during the current study. All

## Abstract

Up to 50% of autistic people experience co-occurring anxiety, which significantly impacts their quality of life. Consequently, developing new interventions (and/ or adapting existing ones) that improve anxiety has been indicated as a priority for clinical research and practice by the autistic community. Despite this, there are very few effective, evidence-based therapies available to autistic people that target anxiety; and those that are available (e.g., autism adapted Cognitive Behavioural Therapy; CBT) can be challenging to access. Thus, the current study will provide an early-stage proof of concept for the feasibility and acceptability of a novel app-based therapeutic approach that has been developed with, and adapted for, autistic people to support them in managing anxiety using UK National Institute for Health and Care Excellence (NICE) recommended adapted CBT approaches. This paper describes the design and methodology of an ethically approved (22/LO/0291) ongoing non-randomised pilot trial that aims to enrol approximately 100 participants aged ≥16-years with an existing autism diagnosis and mild-to-severe self-reported anxiety symptoms (trial registration NCT05302167). Participants will be invited to engage with a self-guided app-based intervention—'Molehill Mountain'. Primary (Generalised Anxiety Disorder Assessment, Hospital Anxiety and Depression Scale) and secondary outcomes (medication/ service use and Goal Attainment Scaling) will be assessed at baseline (Week 2 +/- 2), endpoint (Week 15 +/- 2) and three follow-ups (Weeks 24, 32 and 41 +/- 4). Participants will also be invited to complete an app acceptability survey/ interview at the study endpoint. Analyses will address: 1) app acceptability/ useability and feasibility (via survey/ interview and app usage data); and

relevant data from this study will be made available upon study completion.

**Funding:** This study received funding from the MRC Confidence in Concept award 2019 (1118148) - awarded to BO, ES. DM, ES, BO report grants during the conduct of (but unrelated to) this study from the Innovative Medicines Initiative 2 Joint Undertaking under grant agreement No 777394 for the project AIMS-2-TRIALS. This Joint Undertaking receives support from the European Union's Horizon 2020 research and innovation programme and EFPIA and SFARI, Autistica, Autism Speaks. The views expressed are those of the author(s) and not necessarily those of the NHS, MRC, nor IHI-JU2. The funders had no role in the conceptualisation of this study, nor the development of this publication.

**Competing interests:** The authors have declared that no competing interests exist.

2) target population, performance of outcome measures and ideal timing/ duration of intervention (via primary/ secondary outcome measures and survey/ interview)–with both objectives further informed by a dedicated stakeholder advisory group. The evidence from this study will inform the future optimisation and implementation of Molehill Mountain in a randomised-controlled trial, to provide a novel tool that can be accessed easily by autistic adults and may improve mental health outcomes.

# Introduction

Autism is a neurodevelopmental condition, characterised by core features of social communication/ social imagination difficulties, restricted and repetitive behaviours, and sensory processing differences, with a prevalence of approximately 1% [1, 2]. Up to 50% of autistic people experience clinically relevant co-occurring anxiety by adulthood according to current estimates [3]–though true rates may be higher, given the known diagnostic challenges for identifying anxiety in the autistic population (e.g., diagnostic overshadowing, variability of anxiety presentation, lack of validated autism-specific assessment tools; [4, 5]). This is important, since anxiety can have a significant impact on the lives of autistic people, including contributing to reduced quality of life and difficulties with daily functioning (e.g., at school or work), over and above core autism features [6–9]. Consequently, developing new interventions (and/ or adapting existing ones) that reduce anxiety has been indicated as a key priority for clinical research and practice by the autistic community [10].

However, there are currently very few evidence-based intervention approaches available to support autistic people in managing anxiety [11]. In terms of pharmacological approaches, risperidone and aripiprazole are the only medications approved by regulators (European Medicines Agency, US Food and Drug Administration) for use in autism–but neither target anxiety (they both treat irritability) and both carry risk of unwanted side effects (e.g., fatigue, weight gain; please see [12]). Selective serotonin reuptake inhibitors (SSRIs e.g., sertraline, fluoxetine, citalopram) are specifically recommended to treat anxiety, but evidence for their efficacy is largely based on research in typically developing groups [13]. This is of concern because though pharmacological approaches are effective in reducing anxiety for some autistic people (e.g., see [14]), autistic people are also at heightened risk of adverse side effects (including hyperactivity/ impulsivity, stereotypy, and insomnia; [15]). Furthermore, recent research has shown that the neurological response to serotonin-acting drugs in autistic people is very different to that of non-autistic people [16]. Hence, evidence for the use of current pharmacological anxiety treatments that were developed in neurotypical populations, by autistic people, is very limited.

It is also increasingly acknowledged that social and environmental factors disproportionately experienced by autistic people, such as exposure to societal stigma (and subsequent 'masking' of autism features) and overloading sensory environments, cause anxiety [17–19], and interventions targeted at reducing social inequalities (e.g., Access to Work schemes; [20, 21]) can thus have a supportive effect in reducing anxiety. In parallel, psychological/ psychosocial approaches that incorporate a focus on individual coping strategies for managing some of the social/ environmental contexts implicated in anxiety, such as Cognitive Behavioural Therapy (CBT), are the most widely supported emerging evidence-based therapy for anxiety management in autism [11]. For instance, sensory processing differences, alexithymia (difficulties identifying/ describing emotions), emotion dysregulation, and cognitive biases (e.g.,

intolerance of uncertainty, reduced cognitive flexibility) have all been implicated in anxiety in autism [22–28]; and thus, implementing strategies directly targeting these causal/ maintenance factors may be critical for effective autism-adapted therapeutic approaches.

Meta-analyses assessing the effectiveness of CBT for anxiety in autistic young people and adults have consistently reported moderate-to-large effect sizes for clinician-rated improvement pre- to post-intervention [29–32]. Although these effects are encouraging, there is growing evidence that CBT *needs* to be tailored using autism-specific modifications to be more accessible and effective for autistic people [33]. Modifications may include a more concrete and structured approach to sessions in favour of excessive metaphor and hypotheticals (particularly around identifying and describing emotions), allowing for more processing time through regular breaks, and incorporating autistic strengths like special interests into the therapy to promote engagement [33–36]. These approaches should also acknowledge that the autistic experience (and presentation/ underpinning mechanisms) of anxiety may differ from the non-autistic experience in some respects, including heightened intolerance of uncertainty/ change and anxieties particularly linked to social and/ or sensory stimuli [4, 5, 18]. Indeed, studies directly comparing the impact of adapted vs. standard of practice CBT have identified stronger effects of adapted approaches on anxiety reduction in autistic groups [37].

However, access to adapted CBT for autistic people remains limited in everyday clinical practice. This is partly due to severe pressures on health and social care services (exacerbated by the global COVID-19 pandemic; [38]), which means that resources are predominantly focused on acute mental health provision and so autistic people experiencing long-term mild-to-moderate anxiety often do not meet thresholds to access support (or face long wait times), despite the high impact of anxiety on their daily lives. There is also a lack of available training and support for therapists already delivering adapted CBT approaches [35, 39]. Moreover, many mental health interventions are embedded in traditional 'face to face' clinic services (and increasingly now, telephone appointments) where it is known that autistic people experience barriers to access. These barriers include individual differences in communication needs and feeling misunderstood by healthcare providers–both resulting in feelings of reduced autonomy over care by the service user–as well as difficulties with transitions (e.g., changes in practitioner and/ or service) and sensory difficulties in the clinic environment [40, 41]. Of note, all these factors may be exacerbated by the pre-existing anxiety that the individual is attending clinic services to target (see also [42]).

Novel digital approaches for delivering support and intervention, including app-based technologies, may address some of the challenges outlined above. Additionally, though digital tools should not be viewed as a replacement for in-person services, they may perform an important complementary role, in a way that is: a) (cost)-effective, with app-based interventions having potential to reduce waiting times to early support, including for those awaiting an autism diagnosis or who are self-diagnosed [43]; b) timely, for both the clinic team and service user who can utilise an app-based intervention flexibly at any time and in any place; and c) readily available, with a high proportion of autistic people (especially adolescents and young adults) estimated to have access to a Smart device in the UK context [44]. There is already a wide range of freely available mental health apps that meet Organisation for the Review of Health and Care Apps (ORCHA) digital health quality standards (e.g., SilverCloud; [45]), but none are specifically adapted for autistic people. In fact, we identified just one existing study of an app-based CBT program targeted to supporting autistic people to manage anxiety ('HARU ASD'; available in Korean only), which demonstrated a significant decline in self-reported anxiety for those who used the app over 66 days *(N = 15)*, as compared to a waitlist control group (*N* = 15) in a small community sample [46]. Thus, despite the potential of digital tools for use in mental health service delivery (please see [47]), there is a striking lack of research on the most inclusive, appropriate, and effective methods for implementing these tools [48].

To address this unmet need, the current paper details the protocol for a feasibility study and non-randomised pilot trial of a novel app-based therapeutic approach ('Molehill Mountain') that has been developed with, and adapted for, autistic people to support them in managing anxiety using UK National Institute for Health and Care Excellence (NICE) recommended adapted CBT approaches. The two overarching objectives for this feasibility study are: 1) to establish the acceptability/ useability and feasibility of the Molehill Mountain app in a real-world clinic setting; and 2) to establish the target population, performance of outcome measures and ideal timing/ duration of intervention to inform the design of a future randomised-controlled trial of Molehill Mountain.

## Materials and methods

### Design

This feasibility study employs a non-randomised (single-arm) pilot trial design. The primary location of the study team is King's College London and South London and Maudsley (SLaM) NHS Foundation Trust. SLaM is part of an academic health sciences centre called King's Health Partners with King's College London (KCL), Guy's and St Thomas' (GSTT) and King's College Hospital (KCH).

### Participants

As this is a pilot study, we considered approximately 100 participants to be both a pragmatic (in terms of recruitment/ onboarding rate) and sufficient sample size to examine key feasibility targets—such as, establishing the proportion of those eligible who consent to take part, the proportion of participants who continue to use the app across the 13-week intervention period (and variation in app usage within this), and the proportion who complete each outcome measure at each timepoint (please see [37, 49]).

In further support of the proposed sample size, an indicative *a priori* power analysis using G\*Power version 3.1.9.7 [50] demonstrated that a sample size of $N = 100$ would achieve 80% power to detect a small effect size (0.11) at a significance criterion of $\alpha = .05$ for a repeated measures ANOVA (within subjects). This would allow for the detection of small pre/post effects of intervention on the primary/ secondary outcomes.

Participants will be recruited to this study from outpatient clinic services, existing research databases at King's College London, and public advertising, all in the UK. To be included in the study, participants must have an existing autism diagnosis, be aged 16-years or above (to align with initial community validation of the app in ages 16+ and for the purposes of informed consent provided by participants), be experiencing mild-to-severe anxiety symptoms (as indicated by a score of at least 5 or above on the GAD-7 during screening), be able and willing to provide verbal and written informed consent to take part in the study, be fluent in verbal/ written English for the purposes of engaging with app materials (including with a support person), and be able and willing to use the mobile phone app. Recruitment to the study opened in September 2022 and is currently anticipated to close by June 2023, dependent on rate of participant onboarding. For the schedule of enrolment, interventions, and assessments, please see Fig 1.

Specific exclusion criteria are difficulties with reading/ writing to the extent that the app is inaccessible, high risk of self-harm that make participation in the study inappropriate for the individual's current level of clinical need (as assessed by the clinical team), and having attended $\geq 6$ sessions of therapy (e.g., CBT) in the past 6-months and/ or unstable psychotropic medication use in the past 8-weeks, which would make it impossible to distinguish the effects of the app from existing therapy. Unless otherwise specified above, eligibility to

| | Screen/ enrol | Baseline | Intervention | Endpoint | F/up 1 | F/up 2 | F/up 3 | Time (mins) |
|---|---|---|---|---|---|---|---|---|
| Week | 0 | 2 (+/-2) | 2-15 (+/-2) | 15 (+/-2) | 24 (+/-4) | 32 (+/-4) | 41 (+/-4) | |
| Information sheet | X | | | | | | | - |
| Consent form | X | | | | | | | - |
| Screening (Eligibility check) | X | | | | | | | 20 |
| Demographics | | X | | | | | | 5-10 |
| Exit survey/ interview* | X | | | | | | | 5-10 |
| GAD-7 | X** | | | X | X | X | X | 5-10 |
| HADS | | X | | X | X | X | X | 5-10 |
| Goal Attainment Scaling | | X | | X | X | X | X | 20-30 |
| Medication/ service use questionnaire | | X | | X | X | X | X | 5-10 |
| Autistic traits (ARI, CAT-I) | | X | | | | | | 5-10 |
| App administration | | | X | | | | | 5-10 (daily) |
| Remote contact/ check-in | | | X (Week 8) | | | | | 10-20 |
| App acceptability survey/ interview | | | | X | | | | 30-60 |
| Adverse events | ← | | | | | | → | |

**Fig 1. Participant schedule.** Screen/ enrol and Week 8 remote contact is conducted by the researcher via the participants' preferred mode of communication (e.g., telephone, video call, live chat, email). Baseline, endpoint, follow-up 1–3 assessment (or the exit survey, where applicable) are administered as an online survey via Qualtrics (or the participants' preferred communication method e.g., pen-and-paper, email, telephone). Time in minutes represents the approximate time to complete each assessment per timepoint, however this may be shorter/ longer depending on the individual participant. F/up = Follow-up; GAD-7 = 7-item Generalised Anxiety Disorder Scale; HADS = Hospital Anxiety and Depression Scale; ARI = Adult Routines Inventory; CAT-I = Comprehensive Autistic Trait Inventory. *Only for those who decline/ are not enrolled; **To establish eligibility criteria of mild-to-moderate anxiety symptom severity–also acting as a baseline primary anxiety measure.

participate in the study according to these criteria will be assessed by the research team at study screening (please see further details below). Participants must consent to their GP being notified about their involvement in the study prior to enrolment for safeguarding purposes.

This study is sponsored by King's College London and South London and Maudsley NHS Foundation Trust and funded by the MRC Confidence in Concept scheme. The study has been ethically approved by Bromley Research Ethics Committee (IRAS 308723; 22/LO/0291). Fully informed written consent will be sought from participants by the research team, who will ask participants during enrolment to summarise what the study will involve and their rights to confidentiality and withdrawal to ensure understanding. Model participant consent forms (including easy read versions) are accessible via clinicaltrials.gov (identifier NCT05302167; registered 31st March 2022).

## Intervention

Molehill Mountain (currently Version 2.6 and compatible with both iOS and Android; please see Fig 2 and https://www.autistica.org.uk/molehill-mountain) was developed jointly by King's College London and UK autism charity Autistica, in close working with autistic people, to

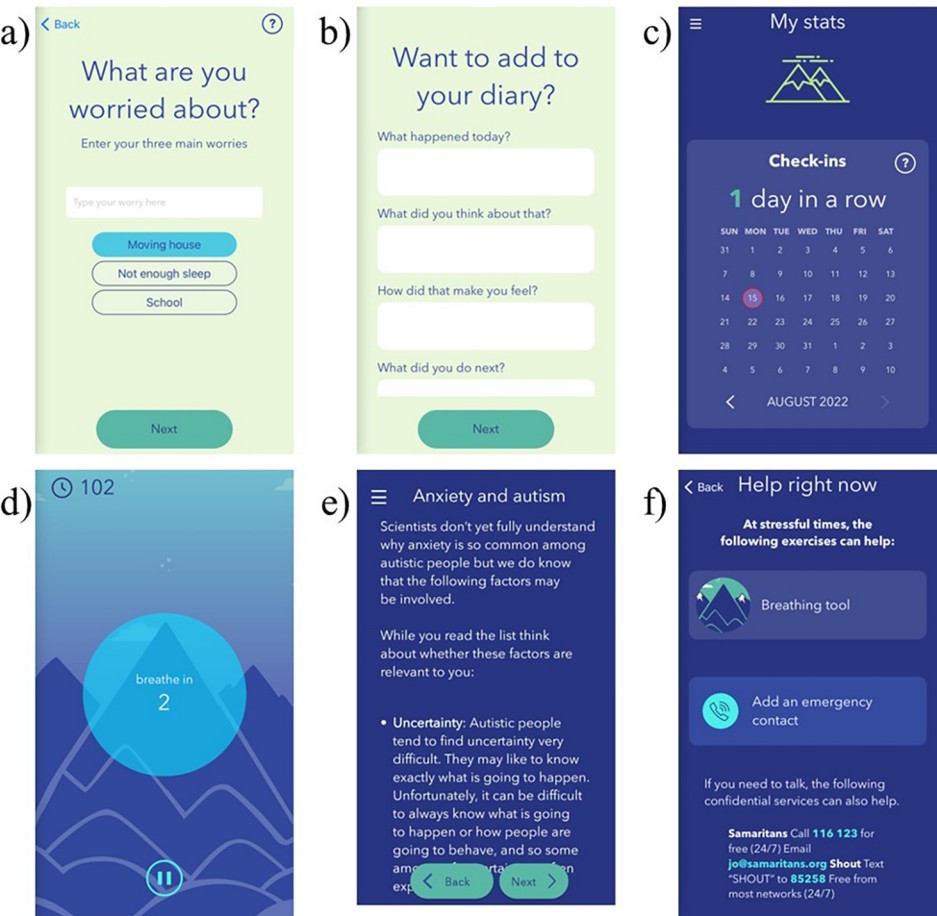

**Fig 2. Molehill Mountain app content and functionality.** A) Daily worries check in; B) Daily diary function; C) App usage and progress statistics; D) Breathing relaxation exercise; E) Anxiety tips and tools incorporating CBT approaches (e.g., cycle of anxiety model, fear ladder); F) Signposting for emergency support. Republished for illustrative purposes from the Molehill Mountain app V2.6 under a CC BY license, with permission from Autistica and King's College London, original copyright 2017.

support anxiety management for those aged 12-years and above using evidence-based adapted CBT principles (please see [37, 51]).

Molehill Mountain consists of six key components to support app users to become more aware of the triggers and protective factors associated with their anxiety and begin to incorporate strategies to manage anxiety. These key components include: 1) reporting on daily anxiety and activities (e.g., how anxious the person has felt today, what they did, and what they have been worried about; Fig 2A); 2) an in-depth daily diary of experiences and emotions (Fig 2B); 3) a progress tracker (e.g., of anxiety over time, changes in worries over time, app usage; Fig 2C); 4) low-intensity anxiety management exercises (e.g., breathing and relaxation techniques; Fig 2D); 5) tips and tools for better understanding and better managing anxiety (based on adapted CBT principles of managing interconnected negative internal thoughts/ emotions, physical sensations and external situations/ actions; Fig 2E); and 6) signposting for emergency support (Fig 2F).

Specific adaptations relevant to autism are embedded in the 'tips and tools' app component, each of which are sequentially unlocked with use of the app over time. These include, for example, psychoeducation on the triggers, presentation, and experiences of anxiety in autistic

people and how these might differ from non-autistic populations, a focus on the physical sensations that may accompany emotions, a focus on alexithymia and emotional literacy, a focus on generalisation of strategies across situations and contexts, and a specific sensory and social anxiety branch. In addition, the mindfulness/ relaxation breathing exercises in the app are accompanied by visual aids.

The Molehill Mountain programme is designed to last up to 3-months, with participants invited to utilise the app daily (i.e., up to once per day, if possible). On first use, app users are prompted to set up a password protected account/ profile to access the app. The daily 'Check In' function allows participants to rate their anxiety level today (from 'Not anxious' to 'Very anxious'), record worries and activities, and add to their daily diary. Nevertheless, the intervention is self-guided by the participant, who will have complete autonomy over when, where, and for how long they engage with the app–also representing a key indicator of app acceptability.

## Outcome measures

**Primary outcome.** The primary outcome in this study will be anxiety symptom severity, as assessed using the self-report 7-item Generalised Anxiety Disorder Assessment (GAD-7; [52]); and Hospital Anxiety and Depression Scale (HADS; [53]). The GAD-7 and HADS were chosen as they are two widely used anxiety assessment tools in UK clinic services for the target population.

The GAD-7 has been reported to be reliable and valid in the general population, with excellent internal consistency (Cronbach's $\alpha$ = 0.92), good test-retest reliability (intraclass correlation = 0.83) and maximal sensitivity/ specificity at a cut-off score of 10 or above–higher scores representing higher anxiety severity [52]. The HADS has also been shown to be reliable and valid for anxiety (and depression) assessment, including in autistic populations where recent reports suggest good internal consistency (Cronbach's $\alpha$ = 0.83) and convergence with other validated measures of anxiety–again, with higher scores representing more anxiety [54].

**Secondary outcome.** The secondary outcomes (also assessed at baseline, endpoint, follow ups 1–3) include individualised functional outcome, as assessed via self-report Goal Attainment Scaling (GAS; S1 Material in S1 File; see also [55]) and assessment of medication/ service use.

The GAS utilised in this study was developed with the Molehill Mountain Advisory Group, whereby common anxieties associated with autism were discussed, formulated as personal goal statements, and then refined down to a final list of 20 goal statements (from an initial list of 38). Thus, GAS allows for each participant to select their own personal goals (a maximum of 4 goals in this study) to be achieved over the course of intervention, while the scoring of the importance of, and progress toward, each goal is standardised to support statistical interpretation. The GAS was included in this study, as its individualised nature and focus on daily functioning may be more sensitive in reflecting subtle (and significant) functional improvements with positive impact on wellbeing in response to intervention than symptom scales like the GAD-7/ HADS. For instance, it is possible that an individual may continue to *feel* anxious, while objectively being more able to engage in everyday activities (e.g., travelling on public transport), as they begin to employ more successful coping strategies for managing those anxious feelings. This is important, since subjective indices of wellbeing in everyday life are known to be meaningful to autistic people yet are a severely underrepresented as outcome measures in intervention research (likely in part due to challenges around standardisation; [56, 57]).

Regarding medication and service use, we ask participants to report whether they are currently taking any medications or dietary supplements (name, dose, start/ end date, purpose), or are involved in any therapies or other interventions (name/ type, hours received, start/ end date, purpose). The purpose of this measure is to both differentiate potential effects of app use

from other interventions and to establish whether app use interacts with other services in any way (e.g., means people use medications/ services more or less at endpoint than before they started using the app).

Lastly, self-reported autistic traits will be integrated into the baseline survey to characterise the sample and for use as a covariate to ascertain whether potential changes in anxiety over the course of intervention are associated with autistic features at baseline. Two existing validated measures of autistic traits are included, as preferred by the Molehill Mountain Advisory Group: the Adult Routines Inventory-Revised (ARI-R; [58]) and the Comprehensive Autistic Trait Inventory (CAT-I; [59]).

**App acceptability/ feasibility.** Last, a main objective of this study is to establish the acceptability and feasibility of the Molehill Mountain app in a real-world clinic setting for future app optimisation. Thus, we will seek participant consent to download app usage data (and participants can further adjust their privacy and sharing preferences directly in the app itself via Settings). In addition, we will administer an in-depth app use experience survey (all participants) and conduct semi-structured qualitative app use experience interviews (subsample of participants). The structure and questions of these measures to gauge app user experience are being designed with the Molehill Mountain Advisory group.

Both approaches provide key indicators of 1) the association between variation in app usage and primary/ secondary outcomes (i.e., the impact of frequency/ duration/ nature of app use on outcomes); 2) reflections on the subjective impact of app use on anxiety and everyday functioning; 3) ease of app administration and use; 4) preferred/ non-preferred app features, content and structure; 5) app technical functionality; and 6) feedback to inform the design of a future randomised-controlled trial (e.g., preparedness of participants to be randomised).

App experience surveys/ interviews will be tailored, according to whether: a) the participant was offered the app but declined, to establish reasons for decline; b) the participant started the app but stopped using it after only a brief time, to establish reasons for loss of engagement (which could also include reduction in anxiety i.e., participants may specifically choose to use the app when feeling more anxious); or c) the participant started and continued to use the app, to establish factors associated with continued engagement.

## Procedure

Full study procedures are provided in detail in the study protocol, which is also accessible via clinicaltrials.gov (Version 2, 18.08.2022; please see S2 Material in S1 File) and will be updated in the case of important protocol modifications.

For participants recruited through outpatient clinics, clinic teams will identify prospective participants who have consented to be contacted about research and may be eligible and will send them information about the study. In parallel, the research team will coordinate recruitment externally from the clinic (e.g., via existing mailing lists, public advertisements). Subsequently, prospective participants who express interest in taking part in the study will be invited to contact the research team directly, who will provide further information (including the participant information sheet) and arrange/ conduct eligibility screening with those who express interest in taking part.

Screening will be conducted by the research team via the participants' preferred mode of communication (e.g., telephone, video call, live chat, email). Firstly, the researcher will check whether participants have the capacity to consent to taking part in the research (e.g., "Can you explain to me what you will be doing in this study?"), answer any questions about the study, and confirm that the prospective participant is comfortable to take part in each study component (e.g., filling questionnaires, using the app). Next, the researcher will ask questions relating

to the inclusion/ exclusion criteria, including confirming participant age, existing autism diagnosis, any recent therapy/ medications, access to a device in order to use the app, and current anxiety according to the GAD-7. Eligible participants who wish to take part will be enrolled into the study (Week 0) and sent the consent form to complete and return online (via Qualtrics/ email) or by post. Participants who are not eligible or who decline to take part at this stage will be invited to complete a short exit survey online via Qualtrics (or the participants' preferred communication method e.g., pen-and-paper, email, telephone), or interview (for a randomly selected subsample of participants who consent to this) including questions about demographics and reasons for decline (if applicable).

Enrolled and consented participants will complete a baseline assessment administered by the research team using Qualtrics (or the participants' preferred communication method), including questions about demographics, autistic traits, and primary and secondary outcomes (HADS, GAS, medication/ service use; Week 2 +/- 2). The GAD-7 will have already been collected at screening to ascertain eligibility for the study based on mild-to-severe anxiety symptom severity. Following the baseline assessment, the research team will support participants to download and initiate the Molehill Mountain app.

Next, participants will undergo a 3-month intervention period engaging with the Molehill Mountain app (Weeks 2–15 +/- 2). The research team will check in with participants at 8-weeks to ensure there are no outstanding queries or concerns from participants and monitor engagement and any adverse events.

At the end of the 3-month intervention period, participants will complete an endpoint assessment administered by the research team using Qualtrics (or the participants' preferred communication method), including primary and secondary outcomes (GAD-7, HADS, GAS, medication/ service use; Week 15 +/- 2). At this stage, participants will also be sent a post-study app use experience survey, or interview for a randomly selected subsample of participants who consent. The endpoint survey/ interview was co-designed with the Molehill Mountain Advisory Group and includes questions about app usage and preferences, self-reported impact of the app on anxiety and related experiences, and experiences of taking part in the research study. Participants who discontinue the intervention prior to Week 15 will be asked whether they consent to complete the endpoint assessment early.

The purpose of the live semi-structured interview component, which will be conducted by the research team in person or remotely, is to gather rich detail on the reasons underlying participant responses to the app use experience survey. Interviews will be conducted with a randomly selected subsample of participants who consent to this–initially 10 participants from each subgroup detailed above (participant declined app, participant stopped app after brief time, participant continued with app). If data saturation has not been reached, interviews will continue in multiples of 3 per subgroup until such time saturation is reached, or the sample for that subgroup exceeds the maximally optimal figure of approximately 30 participants (e.g., [60]).

Finally, participants will be invited to take part in three follow-up assessments (Weeks 24, 32 and 41 +/- 4) where primary and secondary outcomes will be reassessed (GAD-7, HADS, GAS, medication/ service use).

The study is designed to maximise accessibility—all parts are designed to be completed remotely as far as possible (e.g., online, by telephone, email, or by post) and preferred communication will be agreed with each participant at enrolment.

## Statistical analysis

**Data quality and management.**    All research data provided by participants will be pseudonymised prior to analyses by the research team to ensure confidentiality. Data quality (also

including format of measure administration, date/ time of administration, confirmation of data completeness, researcher rated data quality and comments, any other relevant study details, participant withdrawal record) will be recorded in the participant electronic Case Report Form (eCRF) and checked by the research team against Standard Operating Procedures (SOPs) and scoring manuals for each measure. In any cases where data are scored manually (e.g., paper-and-pencil questionnaires), the research team will double code and double enter the data.

It is intended that the results of the study will be reported and disseminated at international conferences and in peer-reviewed scientific journals. Fully anonymised/ summary data will be posted in clinical trial registries and/ or open science repositories at the end of the study, only for participants who consent to this. Participants will also be asked in the consent form if they wish to receive a copy of the results, which they may also provide feedback on ('member checking').

**Statistical analysis plan.** A statistical analysis plan (SAP), co-designed with the Molehill Mountain Advisory Group, will be pre-registered prior to undertaking any inferential statistical analysis. All statistical analysis will be run using R. We will express results in effect sizes with 95% confidence intervals for all analyses to support interpretation. To handle cases of missing data, we will impute the (sub)scale score where no more than 20% of the data for that (sub)scale is missing, otherwise we will exclude the data point from analysis.

Results of analyses will be reported according to the Consolidated Standards of Reporting Trials (CONSORT) guidelines extension for non-randomised trial design. Specifically, details of participant flow and recruitment will be reported, including the number of participants declined/ enrolled, receiving the intervention, analysed on primary/ secondary outcome(s), and losses/ exclusions. Important adverse events (AEs) will also be reported in an anonymised fashion (please see 'monitoring and adverse events' section below).

As this is a feasibility study, we will primarily elicit descriptive statistics (e.g., Mean, Standard Deviation, Range of responses) of participant demographics (baseline) and primary/ secondary outcome measures (baseline, endpoint, follow ups 1–3)—also demonstrating stability and/ or change in anxiety and related features over the course of intervention. We will assess the performance of our primary outcome measures (GAD-7/ HADS) and establish the impact of app use on change in our outcome measures using repeated measures ANOVA (within subjects). This will also inform the design (e.g., optimal outcome measure) for a future randomised-controlled trial and establish the optimal timing for endpoint/ follow-up assessments and interpretation of clinically meaningful reduction in anxiety, based on relationships between the GAD-7/ HADS and secondary outcome measures (GAS, medication/ service use).

Relationships between demographic factors (e.g., age, sex/ gender), autistic traits, objective app usage data, and primary/ secondary outcomes will be explored using simple correlations/ Chi-square to ascertain whether any additional covariates should be included in core analyses and/ or whether exploratory subgroup analyses should be performed. If there is sufficient range in the degree of baseline anxiety (e.g., mild vs. moderate/severe anxiety), we will also explore the impact of baseline anxiety on degree of improvement, covarying for other relevant factors. For instance, we predict that those with milder anxiety may benefit more from the app than those with more severe anxiety (due to their higher support needs).

Subjective app use experiences collected through app use experience surveys/ interviews will also be reported, including anonymised direct quotations from participants who consent. For qualitative data from interviews, thematic analysis will be performed via NVivo, using Braun and Clarke's six-step method [61].

## Monitoring and adverse events

Study monitoring and detection of (serious) adverse events (SAEs/ AEs) will take place continuously throughout the pilot trial. Participants and the clinic team will have the opportunity to contact the research team to report an AE/ SAE at any time and standard operating procedures have been established to respond to these reports (please see protocol; S2 Material in S1 File). The research team will also proactively monitor participant responses (e.g., reports of suicidal intent) at baseline, the week 8 check-in, endpoint, and follow-up assessments.

Additionally, the Trial Management Group (TMG)—which consists of the study Chief and Principal Investigators, research and clinic team members/ external advisors, stakeholders, and representatives of Autistica—will monitor and evaluate study progress, workings, and management on a two-monthly basis, with quarterly reports of any (S)AEs.

## Conclusion

This paper describes one of the first ever acceptability/ feasibility study of an active app-based anxiety intervention ('Molehill Mountain'), developed specifically for autistic people using NICE recommended adapted CBT approaches, and employing a participatory research design. This study will provide early-stage proof-of-concept for the preliminary effectiveness of app use on anxiety and related functional outcomes. Furthermore, objective app usage data and subjective app use experience surveys/ interviews will enable us to identify critical areas for further app optimisation–building an evidence-base to inform the design and implementation of a full-scale randomised-controlled trial.

Future trial design will be developed in co-production with the Molehill Mountain stakeholder advisory group to ensure that the work and its outputs responds to the priorities of the autistic community. For instance, while the selected primary outcome measures (GAD-7, HADS) are currently among the most commonly used in primary and secondary mental health services, the extent to which they capture anxiety features as experienced by autistic people remains unclear. This is because already validated anxiety scales were developed with reference to the general population (though see [62] for initial validation results of the first anxiety measure adapted for autistic adults), however existing research suggests that many autistic people present with anxiety features that fall outside of traditional diagnostic criteria for anxiety. These diverse anxiety features include social anxieties without fear of negative social evaluation, specific phobias of so-called 'unusual' focus or of a sensory nature (e.g., a specific fear of people with beards, or hand dryers), anxiety focused on unknown or non-routine events, and anxieties of a ritualistic nature (e.g., all plug switches must be turned off) in the absence of intrusive thoughts or avoidance of a dreaded situation that would be indicative of obsessive-compulsive disorder [5]. Thus, one key area of impact of the Molehill Mountain advisory group will be to establish the suitability of primary outcomes and their potential alternatives (if applicable), including emerging autism specific anxiety measures like the Anxiety Scale for Autism–Adults [62].

Through the approaches outlined here, the ultimate goal of this work is to better understand for whom an app-based support will most likely be most effective, as a first step to providing more personalised, evidence-based digital therapeutic tools to support some autistic people (particularly those who may find it most challenging to access existing in-person services) to manage anxiety symptoms.

## Supporting information

**S1 Checklist. SPIRIT 2013 checklist: Recommended items to address in a clinical trial protocol and related documents*.**
(DOC)

**S1 File. S1 Material.** Goal attainment scale (Baseline, Endpoint). **S2 Material.** Study protocol.
(DOCX)

## Acknowledgments

We wish to acknowledge the Molehill Mountain stakeholder advisory group at King's College
London: which includes Saffron Baldoza, Amy Hearn, Colin Larkworthy, Adrian Judd, Marianne Savage, and four other members. Correspondence for the Molehill Mountain stakeholder advisory group should be directed to Dr Bethany Oakley (bethany.oakley@kcl.ac.uk).
We wish to acknowledge Autistica, developers and co-owners of the Molehill Mountain app.
Correspondence for Autistica should be directed to research@autistica.org.uk.

## Author Contributions

**Conceptualization:** Bethany Oakley, Sophie Doswell, Antonia Dittner, Ann Ozsivadjian, Rachel Kent, Declan Murphy, Emily Simonoff.

**Data curation:** Charlotte Boatman.

**Funding acquisition:** Bethany Oakley, Sophie Doswell, Antonia Dittner, Ann Ozsivadjian, Rachel Kent, Emily Simonoff.

**Investigation:** Charlotte Boatman.

**Methodology:** Bethany Oakley, Charlotte Boatman, Ann Ozsivadjian, Adrian Judd, Saffron Baldoza, Amy Hearn, Declan Murphy, Emily Simonoff.

**Project administration:** Bethany Oakley, Charlotte Boatman.

**Resources:** Bethany Oakley, Sophie Doswell, Antonia Dittner, Andrew Clarke, Rachel Kent.

**Software:** Andrew Clarke.

**Supervision:** Bethany Oakley, Emily Simonoff.

**Writing – original draft:** Bethany Oakley, Charlotte Boatman.

**Writing – review & editing:** Bethany Oakley, Charlotte Boatman, Sophie Doswell, Antonia Dittner, Andrew Clarke, Ann Ozsivadjian, Rachel Kent, Adrian Judd, Saffron Baldoza, Amy Hearn, Declan Murphy, Emily Simonoff.

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
