## [Decision Letter · Decision Letter 0]

6 Feb 2023

PONE-D-22-34959Molehill Mountain Feasibility Study: Protocol for a non-randomised pilot trial of a novel app-based anxiety intervention for autistic peoplePLOS ONE

Dear Dr. Oakley,

Thank you for submitting your manuscript to PLOS ONE. After careful consideration, we feel that it has merit but does not fully meet PLOS ONE’s publication criteria as it currently stands. Therefore, we invite you to submit a revised version of the manuscript that addresses the points raised during the review process.

We look forward to receiving your revised manuscript.

Kind regards,

Cho Lee Wong, PhD

Academic Editor

PLOS ONE

Journal Requirements:

"This study received funding from the MRC Confidence in Concept award 2019 (1118148). Dr Oakley, Professor Simonoff and Professor Murphy report grants during the conduct of (but unrelated to) this study from the Innovative Medicines Initiative 2 Joint Undertaking under grant agreement No 777394 for the project AIMS-2-TRIALS. This Joint Undertaking receives support from the European Union’s Horizon 2020 research and innovation programme and EFPIA and SFARI, Autistica, Autism Speaks. The views expressed are those of the author(s) and not necessarily those of the NHS, MRC, nor IMI 2JU. The funders had no role in the conceptualisation of this study, nor the development of this publication."

"This study received funding from the MRC Confidence in Concept award 2019 (1118148) - awarded to BO, ES. Dr Oakley, Professor Simonoff and Professor Murphy report grants during the conduct of (but unrelated to) this study from the Innovative Medicines Initiative 2 Joint Undertaking under grant agreement No 777394 for the project AIMS-2-TRIALS. This Joint Undertaking receives support from the European Union’s Horizon 2020 research and innovation programme and EFPIA and SFARI, Autistica, Autism Speaks. The views expressed are those of the author(s) and not necessarily those of the NHS, MRC, nor IMI 2JU. The funders had no role in the conceptualisation of this study, nor the development of this publication. "

5. Please amend the manuscript submission data (via Edit Submission) to include author Judd, Adrian., Baldoza, Saffron., Hearn, Amy.

6. One of the noted authors is a group or consortium The Molehill Mountain Advisory Group. In addition to naming the author group, please list the individual authors and affiliations within this group in the acknowledgments section of your manuscript. Please also indicate clearly a lead author for this group along with a contact email address.

Additional Editor Comments:

Dear Authors,

Thank you for submitting this manuscript for review. The topic is interesting and the protocol addresses a gap in the literature.

However, I have some concerns about the protocol.

Introduction

1) Please elaborate on the main sources/reasons of anxiety in individuals with autism.

Materials and methods

1) The sample size calculation is unclear and there is no reference support. Please explain in details.

2) Like 158: How to determine mild to severe anxiety?

3) Please elaborate on the content and technical aspects of the mobile application (? password required/ Andriod or IOS/ layout).

4) Please detail how the outcome measures will be evaluated (? In-app survey).

Reviewers' comments:

Reviewer's Responses to Questions

**Comments to the Author**

1. Does the manuscript provide a valid rationale for the proposed study, with clearly identified and justified research questions?

Reviewer #1: Partly

Reviewer #2: Partly

2. Is the protocol technically sound and planned in a manner that will lead to a meaningful outcome and allow testing the stated hypotheses?

Reviewer #1: Partly

Reviewer #2: Partly

3. Is the methodology feasible and described in sufficient detail to allow the work to be replicable?

Reviewer #1: Yes

Reviewer #2: No

4. Have the authors described where all data underlying the findings will be made available when the study is complete?

Reviewer #1: Yes

Reviewer #2: No

5. Is the manuscript presented in an intelligible fashion and written in standard English?

Reviewer #1: Yes

Reviewer #2: Yes

6. Review Comments to the Author

You may also provide optional suggestions and comments to authors that they might find helpful in planning their study.

Reviewer #1: The study aims to establish the acceptability/ useability and feasibility of the Molehill Mountain app in a real-world clinic setting; and to establish the target population, performance of outcome measures and ideal timing/ duration of intervention to inform the design of a future randomised-controlled trial of Molehill Mountain.

This study is quite interesting, however, the manuscript requires improvement.

Introduction

Line 139 – 141, the statement ‘To address this unmet need, the current paper details the protocol for a feasibility study and single-arm pilot trial of a novel app-based therapeutic approach (‘Molehill Mountain’) that has been developed with’ requires revision and to align with the title.

Materials and methods

The location to conduct the study, recruitment, and study design employed is to be stated.

Sample size calculation

Line 152, the sample size calculation is unclear and requires more information e.g. to include/state clearly alpha 0.05, one-tailed or two-tailed test, the outcome variable used, and include the word effect size.

Outcome measures

Line 207-244, Line 245-251, Line 250-251, Line 294, Line 309-302, it is not clear how the inventories/questionnaires will be administered. How the inventories/questionnaires (self-report) will be given to subjects and how the participant completed the questionnaires/assessment in the study is to be clearly stated.

It would be good to provide an outline of the final sample questionnaire/survey looks like when incorporating various questionnaires/questions that the participants will be receiving and indicate how long it will take for the participants to complete the questionnaire altogether.

Procedure

Line 273, 278, Line 301, more information on the screening process conducted, how the consent is obtained, how the subjects/participants will be invited and the method of the exit survey will be carried out is to be stated.

Data quality and management

Line 306, full name for CRF to be stated e.g. Case Report Form. The monitoring data quality is to be stated.

Statistical analysis plan

Line 320, 327, 330, 332, the sentence requires revision.

A preliminary possible statistical test could be proposed in the analyses.

Line 344, [50] to be placed after the word method.

Line 346, apart from what were mentioned in the statistical analyses plan, information such as the method to conduct semi-structured interviews, the software to run the statistical analysis and qualitative analysis (if any), the level of accepted statistical significance, correction method (if any), missing data/handling, effect size, 95% CI etc to be stated.

Figure 1, it would be good to add the mode of administration and how the assessment is done for those items highlighted and denoted in the footnote e.g. email, in-person, online etc

Reviewer #2: This is a very exciting topic in a much-needed area of clinical research for this population.

Participants:

• The inclusion criteria are noted to include age, be experiencing anxiety, providing consent, English language fluency, and ability to use a mobile phone app, but there is no information regarding diagnostic status. Earlier in the introduction, it was noted that digital tools can be helpful for those waiting to receive a diagnosis or who are self-diagnosed. How is ASD diagnostic status confirmed or is it by self-report? Is participation based on presence of autistic traits, which are known to be commonly observed in the general population of individuals without an ASD diagnosis (please see Sasson & Bottema-Beutel, 2022 "Studies of autistic traits in the general population are not studies of autism")? As recruitment is through public advertising or outpatient clinic services which may include individuals without an ASD diagnosis, how can the results be interpreted or related to autistic individuals? Could it just be that these ASD-specific modifications and digital app presentation are good for CBT in general and beneficial to anyone, not specifically those with ASD if diagnostic status is not confirmed?

• While the exclusion criteria include things such as recent participation in CBT and recent psychotropic medication use at the start of the study, is there any data being collected from participants as they may start therapy or medication (or other supportive services) while using this app as that may also have an impact on interpretation of results?

Intervention:

• I was unable to find the “evidence-based adapted CBT principles” on the website provided. Please include those references in this manuscript as they provide the critical foundation upon which this app and its specific application for autistic individuals rests and is not provided. It is difficult to evaluate the appropriateness of the modifications and specificity to this population without knowing which evidence-based CBT principles are included in the app or the ASD-specific modifications (and support for them – unless you are referring to those in references 26-29). CBT may have many elements and it is difficult to know which ones are included in the app to evaluate the data and intervention effectiveness.

7. PLOS authors have the option to publish the peer review history of their article (what does this mean?). If published, this will include your full peer review and any attached files.

Reviewer #1: No

Reviewer #2: No

---

## [Author Response · Author response to Decision Letter 0]

31 Mar 2023

Please see Response to Reviewers document attached to this submission.

Response to Journal Requirements

We apologise that some of our manuscript style/ formatting did not initially align with PLOS ONE requirements. We have now compared our manuscript against the PLOS ONE style templates and made the following changes:

• Updating the file names for resubmission

• Provision of short title

• Indicating group authorship and providing author contributions on title pages

• Changes to heading format (Level 1-3)

• Including Figure headings/ captions in main text

• Moving position of acknowledgments

• Adding supporting information section after references

• Removing funding/ competing interests and data availability sections from main text

Thank you for flagging this. We have now updated the Funding Information section accordingly.

"This study received funding from the MRC Confidence in Concept award 2019 (1118148). Dr Oakley, Professor Simonoff and Professor Murphy report grants during the conduct of (but unrelated to) this study from the Innovative Medicines Initiative 2 Joint Undertaking under grant agreement No 777394 for the project AIMS-2-TRIALS. This Joint Undertaking receives support from the European Union’s Horizon 2020 research and innovation programme and EFPIA and SFARI, Autistica, Autism Speaks. The views expressed are those of the author(s) and not necessarily those of the NHS, MRC, nor IMI 2JU. The funders had no role in the conceptualisation of this study, nor the development of this publication."

"This study received funding from the MRC Confidence in Concept award 2019 (1118148) - awarded to BO, ES. Dr Oakley, Professor Simonoff and Professor Murphy report grants during the conduct of (but unrelated to) this study from the Innovative Medicines Initiative 2 Joint Undertaking under grant agreement No 777394 for the project AIMS-2-TRIALS. This Joint Undertaking receives support from the European Union’s Horizon 2020 research and innovation programme and EFPIA and SFARI, Autistica, Autism Speaks. The views expressed are those of the author(s) and not necessarily those of the NHS, MRC, nor IMI 2JU. The funders had no role in the conceptualisation of this study, nor the development of this publication. "

Following journal requirements 1 and 2 (above), we have firstly removed any funding related text from the manuscript. We have also updated the Funding Information section to match the Financial Disclosures text. We wish to maintain the current Funding Statement, with some minor edits, as written below:

“This study received funding from the MRC Confidence in Concept award 2019 (1118148) - awarded to BO, ES. DM, ES, BO report grants during the conduct of (but unrelated to) this study from the Innovative Medicines Initiative 2 Joint Undertaking under grant agreement No 777394 for the project AIMS-2-TRIALS. This Joint Undertaking receives support from the European Union’s Horizon 2020 research and innovation programme and EFPIA and SFARI, Autistica, Autism Speaks. The views expressed are those of the author(s) and not necessarily those of the NHS, MRC, nor IHI-JU2. The funders had no role in the conceptualisation of this study, nor the development of this publication.”

We are committed to making the minimal dataset for this study publicly available, which is also a requirement of our funder. However, as this is a study protocol, we have not yet completed data collection and therefore cannot upload the dataset at this time (neither to the journal, nor repository).

A fully anonymised dataset from this study will be made publicly available after data collection has ended via the clinical trials registry/ open data repository, for those participants who consent to this.

Therefore, as required in the submission portal, we have updated our data availability statement to: “No datasets were generated or analysed during the current study. All relevant data from this study will be made available upon study completion.”

5. Please amend the manuscript submission data (via Edit Submission) to include author Judd, Adrian., Baldoza, Saffron., Hearn, Amy.

One of the noted authors is a group or consortium The Molehill Mountain Advisory Group. In addition to naming the author group, please list the individual authors and affiliations within this group in the acknowledgments section of your manuscript. Please also indicate clearly a lead author for this group along with a contact email address.

We have now added the three additional co-authors to the Manuscript Submission Data, as we received their consent to provide their personal email addresses.

Regarding group authorship, please see Acknowledgements section for individual authors (four members of the group did not provide their consent to be named individually). 

As this is a lived experience expertise group (rather than a consortia), we have indicated the group affiliation as King’s College London, where the group was formed and is hosted, with correspondence to be directed to the lead author of the manuscript whose contact details now clearly stated in the Acknowledgements section.

Response to Editor Comments

1. Introduction 

1) Please elaborate on the main sources/reasons of anxiety in individuals with autism.

We agree that the candidate causal/ maintenance factors associated with anxiety in autism are important to address, given that several of these factors should represent key targets for intervention to reduce anxiety. 

Further evidence is required to elucidate the underpinning mechanisms for anxiety in autism and how they overlap and/or are distinct from those identified in non-autistic populations (acknowledged in new brief addition in brackets on Page 7, Line 118). Nevertheless, existing research would suggest that these mechanisms are multifaceted and include not only genetic/ neurobiological (e.g., autonomic dysregulation due to chronic stress – beyond the scope of this paper, but briefly now referenced on Page 6, Line 98), but also social/ environmental factors (e.g., stigma, social and sensory environmental pressures).

In our resubmitted manuscript, we have particularly focused on adding an overview of factors implicated in anxiety in autism that are also incorporated into the app-based intervention used for this study (and as potentially useful additions to CBT and related approaches more broadly) and thus directly linked to the current research. These include, sensory processing differences, alexithymia (difficulties identifying/ describing emotions), emotion dysregulation, and cognitive biases (e.g., intolerance of uncertainty, reduced cognitive flexibility; please see Page 6, Lines 103-108) – experienced at higher rates in autistic, as compared to non-autistic, populations.

2. Materials and methods

1) The sample size calculation is unclear and there is no reference support. Please explain in details.

As our study is a feasibility study, we based our original sample size on practicality/ feasibility for a 12-month recruitment window, and our best estimate of the available target participant pool (i.e., those on the waitlist) in local mental health services – following guidance on the sample size for pilot study designs as in Leon et al (2011; 10.1016/j.jpsychires.2010.10.008). This was to allow us to address key study feasibility targets, such as proportion who are eligible and consent, proportion who use the app over 13-weeks and complete outcome measures, ease of app administration and use, preferred and non-preferred app features, content, and structure, subjective reflections on the impact of app use on anxiety and everyday wellbeing/ functioning, and preparedness of participants to be randomised in a future trial.

Following the recommendations of the Editors and Reviewers, we now also provide an indicative sample size calculation to further support our approach. We have updated this in the manuscript accordingly (Page 9, Line 175-185).

Reflecting on the combination of guidance on pilot study sample size estimates, and the power calculation provided, we have revised our target sample size to approximately 100 participants.

3. 2) Like 158: How to determine mild to severe anxiety?

We have now specified that we establish this inclusion criteria as indicated by a score of at least 5 or above on the GAD-7 during screening (please see Page 10, Line 191-192) and we provide specifiers for all bands (mild, moderate severe) of the GAD-7 scoring cutoffs (please see Page 13, Lines 273-274) to provide more context on this use of the tool for the reader.

3. 3) Please elaborate on the content and technical aspects of the mobile application (? password required/ Andriod or IOS/ layout).

We agree that it would be helpful to the reader to provide some more detail on the content and technical aspects of the app. We now confirm in the main text of the manuscript that the app is compatible with both iOS and Android (please see Page 11, Line 228), that the app is accessed through a password protected individual user account/ profile (please see Page 12, Lines 257-258), that participants can adjust their privacy and sharing preferences directly in the app under Settings (please see Page 15, Lines 315-316), and more detail about the daily ‘Check In’ function (please see Page 12, Lines 257-260).

Further technical details and development information about the app are provided on the Autistica website (Molehill Mountain | Autistica), which we also reference in the manuscript.

4. 4) Please detail how the outcome measures will be evaluated (? In-app survey).

We clarify this by adding that the baseline/ endpoint and follow up assessments (which incorporate the outcome measures used in this study) are administered by the research team via Qualtrics (please see Page 10, Figure 1 Legend; Page 16 Lines 353-354 and 355-356, 360; Page 17 Line 371), or the participants’ preferred communication method to enhance inclusivity of the study design (e.g., via email, post, over the telephone).

Response to Reviewer Comments

Reviewer #1 Comments and Author Response: 

1. Introduction

Line 139 – 141, the statement ‘To address this unmet need, the current paper details the protocol for a feasibility study and single-arm pilot trial of a novel app-based therapeutic approach (‘Molehill Mountain’) that has been developed with’ requires revision and to align with the title.

We thank the reviewer for flagging this discrepancy and have updated this statement in the Introduction accordingly to align with the title (please see Page 8, Lines 158-159).

2. Materials and methods

The location to conduct the study, recruitment, and study design employed is to be stated.

Following the reviewers’ recommendation, we have added a new ‘Design’ section to the Materials and methods (please see Page 9, Lines 169-173), which also details the primary study location. Further details are also provided regarding recruitment locations on Page 9, Lines 186-187.

3. Sample size calculation

Line 152, the sample size calculation is unclear and requires more information e.g. to include/state clearly alpha 0.05, one-tailed or two-tailed test, the outcome variable used, and include the word effect size.

We apologise for the lack of clarity on the sample size previously provided and have updated this section in the manuscript accordingly (Page 9, Line 175-185). For a full explanation, please see response to Editor Comments, comment and response #2, above.

4. Outcome measures

Line 207-244, Line 245-251, Line 250-251, Line 294, Line 309-302, it is not clear how the inventories/questionnaires will be administered. How the inventories/questionnaires (self-report) will be given to subjects and how the participant completed the questionnaires/assessment in the study is to be clearly stated.

As also noted in response to Editor Comments, comment and response #4, we now clarify this by adding that the baseline/ endpoint and follow up assessments (which incorporate the outcome measures used in this study) are administered by the research team via Qualtrics (please see Page 10, Figure 1 Legend; Page 16 Lines 353-354 and 355-356, 360; Page 17 Line 371), or the participants’ preferred communication method to enhance inclusivity of the study design (e.g., via email, post, over the telephone).

5. It would be good to provide an outline of the final sample questionnaire/survey looks like when incorporating various questionnaires/questions that the participants will be receiving and indicate how long it will take for the participants to complete the questionnaire altogether.

We thank the reviewer for the suggestion to provide an indication of how long it will take to complete the survey components of the study and we have now added this information into Figure 1, where we provide a full summary of which questionnaires are included per survey/ timepoint. 

6. Procedure

Line 273, 278, Line 301, more information on the screening process conducted, how the consent is obtained, how the subjects/participants will be invited and the method of the exit survey will be carried out is to be stated.

We agree with the reviewer that more details and context would be helpful for the reader in the Procedures section and make the order of events in the study clearer. We therefore provide additional information regarding inviting participants (please see Page 16, Lines 336-338, 340-343), the screening process (please see Page 16, Lines 344-352), obtaining consent (please see Page 16, Lines 352-354), and exit survey administration (please see Page 16, Lines 354-358 and Figure 1 legend).

7. Data quality and management

Line 306, full name for CRF to be stated e.g. Case Report Form. The monitoring data quality is to be stated.

We apologise for this oversight and have now corrected the acronym to the full term (please see Page 18, Line 401) and further detail added to this section (Page 18, Lines 398-401).

8. Statistical analysis plan

Line 320, 327, 330, 332, the sentence requires revision.

We have followed the above recommendation of the reviewer by:

• Clarifying that results with be reported according to CONSORT extension for non randomised trial design (Page 19, Lines 418-419).

• Clarifying from which timepoints descriptive statistics will be reported on which measures (Page 19, Lines 425-428).

• Clarifying that performance of outcome measures will inform the design (e.g., in terms of optimal assessments) for a future RCT (please see Page 428-435) and interpretation of changes in anxiety that may be meaningful (e.g., because they relate to other functional outcomes).

A preliminary possible statistical test could be proposed in the analyses.

We agree with the reviewer that we should be more specific in this regard and have added some relevant examples (please see Page 19, Lines 425-426, 430-431, 437-438).

Line 344, [50] to be placed after the word method.

We thank the reviewer for noticing this typo, and have updated accordingly (Page 20, Line 447).

Line 346, apart from what were mentioned in the statistical analyses plan, information such as the method to conduct semi-structured interviews, the software to run the statistical analysis and qualitative analysis (if any), the level of accepted statistical significance, correction method (if any), missing data/handling, effect size, 95% CI etc to be stated.

We have added further information on these aspects in the resubmitted manuscript, including our planned approach to statistical reporting and interpretation (please see Page 19, Lines 413-417), the software to be used (please see Page 19, Lines 413-414; Page 20, Line 447). We also provide a more detailed rationale for the app experience interview component on Page 17, Lines 375-388.

9. Figure 1, it would be good to add the mode of administration and how the assessment is done for those items highlighted and denoted in the footnote e.g. email, in-person, online etc

We now indicate the mode of administration for each assessment in the Figure legend (please see Page 10), also complementing our edits in response to Editor (comment and response #4) and Reviewer #1 (comment and response #4) queries regarding how the questionnaires are administered.

Reviewer #2 Comments and Author Response: 

1. Participants:

• The inclusion criteria are noted to include age, be experiencing anxiety, providing consent, English language fluency, and ability to use a mobile phone app, but there is no information regarding diagnostic status. Earlier in the introduction, it was noted that digital tools can be helpful for those waiting to receive a diagnosis or who are self-diagnosed. How is ASD diagnostic status confirmed or is it by self-report? Is participation based on presence of autistic traits, which are known to be commonly observed in the general population of individuals without an ASD diagnosis (please see Sasson & Bottema-Beutel, 2022 "Studies of autistic traits in the general population are not studies of autism")? As recruitment is through public advertising or outpatient clinic services which may include individuals without an ASD diagnosis, how can the results be interpreted or related to autistic individuals? Could it just be that these ASD-specific modifications and digital app presentation are good for CBT in general and beneficial to anyone, not specifically those with ASD if diagnostic status is not confirmed?

We apologise that we were not clearer on this point and agree with the reviewer that this is an important clarification for us to make. We confirm that we do require participants to have an existing autism diagnosis to take part in the study. We had previously included reference to an existing diagnosis of autism along with information on the recruitment approach, which meant that this inclusion criteria was not obvious in our original submission. Therefore, we have now stated as part of the inclusion criteria that an autism diagnosis is required to take part in this study (please see Page 9, Line 188).

We ascertain this information in two ways. For participants recruited through clinic services, the clinic team screen for potentially eligible participants and thus do not include those without an existing autism diagnosis (as per existing clinic records) in this process. For all participants, and particularly relevant to those recruited outside of clinic services, the research team also request confirmation of autism diagnosis (including the specific diagnosis if known e.g., ASD, Asperger’s, and who gave this diagnosis) during the screening process and we have added this information in brief to Page 16, Line 350-351.

In terms of the later interpretation of findings, we also include two dimensional measures of autistic traits in our assessment battery (CAT-I, ARI) to enable us to ascertain whether baseline autistic traits influence app usage/ its impact on anxiety outcomes across the whole sample to identify potential individual variation in app acceptability/ effectiveness in autism.

2. While the exclusion criteria include things such as recent participation in CBT and recent psychotropic medication use at the start of the study, is there any data being collected from participants as they may start therapy or medication (or other supportive services) while using this app as that may also have an impact on interpretation of results?

We thank the reviewer for highlighting that we had inadvertently not provided details of our questionnaire on medication/ service use (secondary outcome measure) in the methods, which we now provide on Page 14, Lines 299-305.

3. Intervention:

• I was unable to find the “evidence-based adapted CBT principles” on the website provided. Please include those references in this manuscript as they provide the critical foundation upon which this app and its specific application for autistic individuals rests and is not provided. It is difficult to evaluate the appropriateness of the modifications and specificity to this population without knowing which evidence-based CBT principles are included in the app or the ASD-specific modifications (and support for them – unless you are referring to those in references 26-29). CBT may have many elements and it is difficult to know which ones are included in the app to evaluate the data and intervention effectiveness.

We agree with the reviewer that the evidence-base for the features of the app are core to the study approach and interpretation. We have therefore added further detail about specific app components and their underlying evidence base on Page 12, Lines 248-255 and provide additional references, including from our team, to support this (please see Chew, Ozsivadjian, Hollocks and Magiati, 2022; Wood et al, 2020).

---

## [Editor Report · Decision Letter 1]

24 May 2023

Molehill Mountain Feasibility Study: Protocol for a non-randomised pilot trial of a novel app-based anxiety intervention for autistic people

PONE-D-22-34959R1

Dear Dr. Oakley

We’re pleased to inform you that your manuscript has been judged scientifically suitable for publication and will be formally accepted for publication once it meets all outstanding technical requirements.

Kind regards,

Cho Lee Wong, PhD

Academic Editor

PLOS ONE
---

## [Editor Report · Acceptance letter]

22 Jun 2023

PONE-D-22-34959R1 

Molehill Mountain feasibility study: Protocol for a non-randomised pilot trial of a novel app-based anxiety intervention for autistic people 

Dear Dr. Oakley:

I'm pleased to inform you that your manuscript has been deemed suitable for publication in PLOS ONE. Congratulations! Your manuscript is now with our production department. 

Kind regards, 

on behalf of

Dr. Cho Lee Wong 

Academic Editor

PLOS ONE